# Reverse Transcription Recombinase Polymerase Amplification Assay for Rapid Detection of Avian Influenza Virus H9N2 HA Gene

**DOI:** 10.3390/vetsci8070134

**Published:** 2021-07-16

**Authors:** Nahed Yehia, Fatma Eldemery, Abdel-Satar Arafa, Ahmed Abd El Wahed, Ahmed El Sanousi, Manfred Weidmann, Mohamed Shalaby

**Affiliations:** 1National Laboratory for Quality Control on Poultry Production, Animal Health Research Institute, Agriculture Research Center, Dokki, Giza 12618, Egypt; nahedyehia@gmail.com (N.Y.); araby85@hotmail.com (A.-S.A.); 2Department of Hygiene and Zoonoses, Faculty of Veterinary Medicine, Mansoura University, Mansoura 35516, Egypt; fatmaelbaz@mans.edu.eg; 3Division of Microbiology and Animal Hygiene, Faculty of Agricultural Sciences, University of Goettingen, 7077 Goettingen, Germany; 4Institute of Animal Hygiene and Veterinary Public Health, Faculty of Veterinary Medicine, University of Leipzig, 04103 Leipzig, Germany; 5Department of Virology, Faculty of Veterinary Medicine, Cairo University, Cairo 12211, Egypt; sanousi.a@cu.edu.eg (A.E.S.); mshalaby43@gmail.com (M.S.); 6Institute of Microbiology & Virology, Brandenburg Medical School, 01968 Senftenberg, Germany; Manfred.Weidmann@mhb-fontane.de

**Keywords:** avian influenza, H9N2, RT-RPA, RT-PCR, diagnosis

## Abstract

The H9N2 subtype of avian influenza A virus (aIAV) is circulating among birds worldwide, leading to severe economic losses. H9N2 cocirculation with other highly pathogenic aIAVs has the potential to contribute to the rise of new strains with pandemic potential. Therefore, rapid detection of H9 aIAVs infection is crucial to control virus spread. A qualitative reverse transcription recombinase polymerase amplification (RT-RPA) assay for the detection of aIAV subtype H9N2 was developed. All results were compared to the gold standard (real-time reverse transcription polymerase chain reaction (RT-PCR)). The RT-RPA assay was designed to detect the hemagglutinin (HA) gene of H9N2 by testing three pairs of primers and a probe. A serial concentration between 10^6^ and 10^0^ EID_50_ (50% embryo infective dose)/mL was applied to calculate the analytical sensitivity. The H9 RT-RPA assay was highly sensitive as the lowest concentration point of a standard range at one EID_50_/mL was detected after 5 to 8 min. The H9N2 RT-RPA assay was highly specific as nucleic acid extracted from H9 negative samples and from other avian pathogens were not cross detected. The diagnostic sensitivity when testing clinical samples was 100% for RT-RPA and RT-PCR. In conclusion, H9N2 RT-RPA is a rapid sensitive and specific assay that easily operable in a portable device for field diagnosis of aIAV H9N2.

## 1. Introduction

Influenza A viruses (IAVs) are enveloped, single-stranded, negative-sense RNA viruses [1]. The IAV genome is segmented into eight RNA segments: RNA polymerase basic subunit (PB)2, PB1, RNA polymerase acidic subunit (PA), hemagglutinin (HA), nucleoprotein (NP), neuraminidase (NA), matrix (M) (encodes two matrix proteins (M1 and M2)), and non-structural (NS) segment [2]. Influenza A viruses are divided into subtypes based on HA and NA. So far, 18 HA subtypes and 11 NA subtypes have been identified thus, many different combinations of HA and NA exist [3]. H9N2 aIAVs continue to evolve and circulate in different bird species and poultry, and pose a public health concern [4]. The crucial characteristics of aIAVs in general and H9N2 virus in particular are the wide host range, extensive gene reassortment, and adaptation to wild birds, poultry and mammals [5,6]. Co-circulation of H9N2 with other highly pathogenic aIAVs such as H5N1 and H5N8 has been reported in Egypt [7,8,9]. For instance, recent analyses of human infections with aIAVs showed that H9N2 is the gene donor for H7N9 and H10N8 viruses [10], and that H9N2 has a great potential to emerge as a new virus infecting humans [4,11]. Additionally, the Egyptian H9N2 viruses were found to bind more to human-like receptors (α2,6-SL, Neu5Acα2-6Galβ1-4Glc, α2,6-SLN, Neu5Acα2-6Galβ1-4GlcNAc) than to avian-like receptors (α2,3-SL, Neu5Acα2-3Galβ1-4Glc) [12].

H9N2 has emerged and become endemic in Egypt since 2010, with a serious economic impact on the poultry industry [9]. When H9N2 infection occurs combined with secondary infections, high morbidity and mortality and great economic losses are recorded [13,14]. H9N2 subtypes circulating from 2016–2018 presented a mix of more recent segments emerging since 2014 and older M1, M2, and NS gene sequences known from H9N2 virus isolates of 2010, indicating ongoing drift and reassortment amongst H9N2 viruses [7]. However, the Egyptian H9N2 viruses are low pathogenic (LPAI), asymptomatically infected poultry shed and spread the virus via the oral and cloacal route sustaining circulation. This silent spread of the virus in the field might generate new sub- and genotypes of aIAVs. Recently more antigenic drifts were recorded for Egyptian H9N2 isolates compared to isolates from other countries [7]. Moreover, the NA enzymatic activities of Egyptian N9N2 strains were tremendous [15]. Thus, Egypt is potentially developing into a hotspot for new influenza pandemics [16].

Rapid point-of-need diagnostic tests are necessary in order to stop the viral spread and decrease the burden on the economy [17]. Routine detection of aIAVs in clinical samples is performed by viral isolation, which is time-consuming. DNA amplification-based molecular techniques such as polymerase chain reaction (RT-PCR) are more sensitive and rapid but need sophisticated equipment and protocols [18]. In addition, the RT-PCR assay has a long runtime of more than one hour [19,20]. The recombinase polymerase amplification (RPA) technique has been successfully developed as a rapid diagnostic method of aIAVs viruses such as H7N9 and H5N1 [19,21,22,23,24]. Unlike PCR, RPA works at a constant low temperature and completes amplification within minutes [25]. The reaction is accomplished by a particular enzyme mixture instead of seven proteins and repetitive thermal cycling. The recombinase, strand-displacing DNA polymerase and single-stranded DNA-binding protein represent the core of the RPA reaction at temperatures between 37 °C and 42 °C. The RPA product can be visualized using electrophoresis, lateral flow dipsticks (LFD) [22,24], or by probe-based real-time amplification generating results in less than 20 min [19,26]. In this study, we aimed to develop a probe-based RT-RPA and evaluated its performance for field detection of the HA gene of avian influenza H9N2.

## 2. Materials and Methods

### 2.1. Ethic Statement

All samples used in the study were part of the routine national surveillance program of IAVs in Egypt, and, therefore, no ethical approval was required. No sample was collected, or animals were slaughtered specifically for this study because all samples were archived RNA samples from the national depository at the Egyptian National Reference Laboratory for IAVs (Animal Health Research Institute, Agriculture Research Centre, AHRI, Giza, Egypt).

### 2.2. Generation of RNA Standard

The standard RNA was extracted from Egyptian H9N2 reference strain (A/chicken/Egypt/1373Vd/2013(H9N2)), GenBank accession number KJ781216 that was titrated using specific pathogen free (SPF) embryonated chicken eggs (ECE). Virus titer shown as EID_50_ (50% embryo infective dose)/mL was calculated using the Reed and Muench method, as previously described [27].

### 2.3. Viral RNA Extraction

The QIAamp Viral RNA Mini Kit was applied according to the manufacturer’s directions (Qiagen, Hilden, Germany) to extract the viral RNA from clinical swabs. Samples were either allantoic fluid or phosphate-buffered saline (PBS) suspensions containing tracheal swabs of a total volume of 200 µL. RNA was eluted in a final volume of 60 µL and stored at −80 °C.

### 2.4. H9 RT-RPA Primers and Exo Probe

Three forward, three reverse primers, and one exo-probe were used to amplify ~240–280 pb of the HA gene to select the best combination providing the highest RT-RPA assay sensitivity (Table 1). A total of 31 HA gene sequences of Egyptian H9N2 isolates from 2010–2012 were used to design the oligonucleotides (GenBank accession numbers: JX192599.1, KJ781208.1, KJ781213.1, KJ781214.1, KJ781215.1, KJ781210.1, KJ781211.1, KJ781212.1, JX192601.1, KJ781207.1, KJ781209.1, JQ906552.1, JQ440373.2, JX192600.1, JQ906554.1, JQ906555.1, JQ906557.1, JQ906558.1, JQ906559.1, JQ906560.1, JQ906556.1, KC017474.1, JX912997.1, CY110926.1, KF881651.1, JX273139.1, CY126239.1, JN828572.1, GQ120549, JQ419502.2, KJ781216). The selection of the conserved region was performed by multiple sequence alignment in the MegAlign program (DNASTAR Inc., Madison, WI, USA). All selected RPA primers and exo-probes were designed in accordance with the Twist Amp exo RT kits manual (Twist Dx, Cambridge, UK) and ordered from Tib MolBiol (Berlin, Germany).

### 2.5. Optimization of H9 RT-RPA Conditions

The H9 RT-RPA was performed in a 50 µL reaction volume using Twist Amp exo RT kits (Twist Dx, Cambridge, UK) according to the manufacturer’s directions, as previously described [19]. In brief, the 50 µL reaction was performed according to the following formula: 1 µL of each RPA primers (10 pmol Conc.), 0.6 µL RPA exo-probe (10 pmol Conc.), 4 µL of magnesium acetate (14 mM Conc.), 29.5 µL of 4 × rehydration buffer (Twist Amp, Cambridge, UK), and 1 µL RNA template then PCR grade water (Qiagen, Hilden, Germany) was added to adjust the reaction volume up to 50 µL. This mix was added to the RPA strips containing a dried enzyme pellet. Heating and fluorescence measurements were conducted in an ESE Quant tube scanner (Qiagen, Lake Constance, Germany) at 42 °C for 20 min. The threshold time (TT) was determined by combining threshold and signals loop analysis confirmed by 1st derivative analysis in the tube scanner software.

### 2.6. H9 RT-RPA Analytical Sensitivity

The limit of detection of H9 RT-RPA assay was determined using a serial dilution range of titrated aIAVs EID_50_ from 10^5^ EID_50_/mL to 10 EID_50_/mL in eight replicates. The RNA concentration was calculated using the previously developed real-time RT-PCR [28] and Quantitect probe RT-PCR kit (Qiagen, Inc., Valencia, CA, USA). PRISM (Graphpad Software Inc. version 6.0 Mac, SanDiego, CA, USA) was applied to plot the RT-RPA threshold time against RNA molecules detected after performing semi-log regression.

### 2.7. H9 RT-RPA Analytical Specificity

Nucleic acid from other influenza viruses, including H5N2 and H7N1, and three viruses producing respiratory signs including infectious laryngotracheitis virus (ILTV), infectious bronchitis virus (IBV) and Newcastle disease virus (NDV) (supplemented from GD Lab, Holland, a regional lab of OIE lab), *Mycoplasma gallisepticum* (Cornell University Diagnostic Lab., New York, NY, USA), and six H9 real-time RT-PCR negative samples from apparently healthy chickens were used to determine the assay cross-reactivities.

### 2.8. Clinical Accuracy of H9 RT-RPA

Thirty tracheal swabs were collected from field cases in Egypt and twelve negative tracheal swabs from specific pathogen-free (SPF) chicks and apparently healthy chickens were used to test the clinical performance of H9 RT-RPA assay compared to real-time RT-PCR. The samples varied from strong cycle threshold 13 (13 CT) to very weak sensitive samples (>35 CT). The real-time RT-PCR was performed using a Quantitect probe RT-PCR kit (Qiagen, Valencia, CA, USA) and using previously published oligonucleotides [28]. Linear regression analysis using PRISM (Graphpad Software Inc. version 6.0 Mac, San Diego, CA, USA) were applied to determine the correlation between real-time RT-PCR CT values and H9 RT-RPA TT.

### 2.9. Statistical Methods

The RT-RPA TT was determined using the Tube Scanner Software (Qiagen, Lake Constance, Germany) as the first derivatives of the measured fluorescence signal in real-time. The value of the negative control was used as a cut-off value. The clinical sensitivity and specificity of the RT-RPA were determined using the standard formula published previously [29]. Linear regression analysis was performed using PRISM (Graphpad Software Inc. version 6.0 Mac, San Diego, CA, USA) by considering each Y value as an individual point with a 95% confidence interval value.

## 3. Results

### 3.1. Screening of the Primer Sets

An efficient primer set is crucial for the optimization of the RT-RPA assay. Three primers/probe combinations targeting the HA2 region of H9 (A/chicken/Egypt/1373Vd/2013(H9N2) were screened for their ability to amplify 240–280 pb of 10^5^ EID_50_ per RT-RPA reaction mixture (Table 1). Primer pair H9 RPA-F1+ H9 RPA-R1 yielded the best amplification with signals emerging at 5.3 min among all the primer combinations (Figure 1A). The fluorescence amplitude of H9 RPA-F1+ H9 RPA-R1 was consistently the highest among all the primer pairs tested. Thus, this primer set (H9 RPA-F1+ H9 RPA-R1) was used to establish the RT-RPA assay and used for further assay validation.

### 3.2. Sensitivity of H9 RT-RPA Assay

To determine the analytical sensitivity of H9 RT-RPA assay, a dilution range of 10^5^-10^0^ molecules/µL of the H9 RNA standard was used (Figure 1B). When testing the 10 EID_50_ range primer H9 RPA-F1+ H9 RPA-R1 also detected 10 EID_50_/mL in 8 min. The sensitivity of the real-time RT-PCR was 10^10^ EID_50_/mL. Semi-logarithmic regression of the data collected from eight H9 RT-RPA test runs on the RNA standard using PRISM. It yielded results between 5–8 min (Figure 2) and probit regression was not applied because the H9 RT-RPA detected eight times out of eight in all dilutions.

### 3.3. Specificity of H9 RT-RPA Assay

When testing the specificity of H9 RT-RPA against other poultry pathogens producing respiratory manifestations, including aIAVs H5N2, H7N1, ILTV, IBV, NDV, and mycoplasma, only H9N2 was detected, indicating that the primers H9 RPA-F1+ H9 RPA-R1 were highly specific (Figure 3). In addition, no unspecific amplification was detected in control negative samples from SPF and apparently healthy chickens.

### 3.4. Clinical Accuracy of H9 RT-RPA Assay

Validation of the RT-RPA assay compared to RT-PCR assay was evaluated in thirty clinical samples. The results of H9 RT-RPA were detected in about 3–7 min in high and low virus load samples. The sensitivity of both the H9 RT-RPA and real-time RT-PCR assays was 100%. Linear regression analysis of H9 RT-RPA threshold time and PCR-cycle threshold was performed (Figure 4), yielding a R square value of 0.12. This weak relationship between H9 RT-RPA and real-time RT-PCR is due to the speed of the RPA assay.

## 4. Discussion

Avian influenza virus subtype H9N2 is prevalent in Egypt and represents a large portion of the avian influenza subtypes detected among domestic poultry. Rapid field diagnosis of H9N2 subtype aIAVs is essential to reduce virus spread, economic losses, and human infections. However, real-time RT-PCR is the method of choice for the detection of avian influenza [28]. It is difficult to use in field conditions because of complex and costly instruments. Therefore, in this study we developed RT-RPA assay for rapid, sensitive, and specific on-site detection of the HA gene.

Primer design in the RPA assay is still a big bottleneck, as apart from some guidelines, rules are not described and automated software design is, as yet, impossible [25]. Therefore, several primes were tested before selecting the best primer combination. Nine possible combinations were screened for sensitive amplification of the H9N2 viral RNA standard. Only the H9 RPA-F1+ H9 RPA-R1 primers produced a limit of detection of one EID_50/_mL.

The H9N2 RT-RPA assay is a highly specific assay because only the viral RNA of H9N2 viruses were amplified. The nucleic acid of other respiratory pathogens in poultry, as well as specific pathogen-free healthy chickens, were scored negative in the H9 RT-RPA assay. The oligonucleotides of the developed assay were placed in a very conserved and specific region as done previously in the real-time RT-PCR [28]. The H9N2 RT-RPA assay correctly detected all real-time RT-PCR positive chicken samples (*n* = 30). Many samples with very high CT values around 38 in real-time RT-PCR were identified as positive by RT-RPA in less than 7 min, which is an indication of the high sensitivity of the developed assay. A correlation between the CT values of the real-time RT-PCR and TT values of the RT-RPA was not observed due to the explosive nature of the RPA reaction, which outruns the repetitive, regular amplification cycles of the PCR reaction [30].

Another promising isothermal application is the loop-mediated isothermal amplification (RT-LAMP) assay [31]. Compared to LAMP, the H9 RT-RPA is faster (less than 10 min in H9 RT-RPA, while at least 30 min is needed in RT-LAMP). Moreover, the number of primers is two in RPA and four in RT-LAMP [31]. Recently, RT-RPA-combined lateral flow dipstick (FLD) assay was developed for the detection of the H9 gene [24]. Compared to RT-RPA-FLD assay, our probe-based RPA assay is fast (7 min) because it can generate real-time amplification results in less than 20 min without an additional process after amplification. In addition, the clinical performance of H9 RT-RPA is slightly better than H9 RT-RPA-FLD as the agreement of the detection results between RT-RPA-LFD and conventional RT-PCR was 95.8% [24]. Recently, insulated isothermal PCR (iiPCR) devices were developed and assessed for the detection of avian influenza. The field-deployable POCKIT Micro DUO Nucleic Acid Analyzer needs 45 min, while the POCKIT Central Nucleic Acid Analyser processes results in 85 min to perform an iiPCR analysis [32]. In terms of time to result, both devices are still not as fast as a probe-based RPA assay under field conditions.

Recent phylogenetic analyses show that H9N2 can be classified into three main lineages, each showing lineage-specific evolution. For the Egyptian strains, the HA segment sequences targeted by the assay described here form a distinctive subclade in the Asian-African lineage [33]. The assay designed specifically for this subclade should therefore not be used for the detection of other subclades. RPA primers are known to tolerate up to 9 mismatches in the target sequence [34], and indeed, in some of our own work, RPA was shown to be very robust in detecting variant FMDV genotypes despite a considerable number of mismatches between genotypes [20]. Nevertheless, regular sequencing of local strains and alignment with the amplicon, as well as testing the viral RNA of evolving H9N2 strains, will be crucial to keep the presented RT-RPA amplicon up to date to avoid false negatives in the future.

In conclusion, the developed H9 RT-RPA is rapid, simple, and sensitive for the on-site detection of H9N2 aIAVs. We have previously designed a “diagnostics-in-a-suitcase” for several pathogens [20,30,35], which can be applied for rapid detection of the virus under field settings, in order to start immediate control measures, nevertheless, according to national Egyptian law, all positive samples must be confirmed by governmental institutions.

## Figures and Tables

**Figure 1 vetsci-08-00134-f001:**
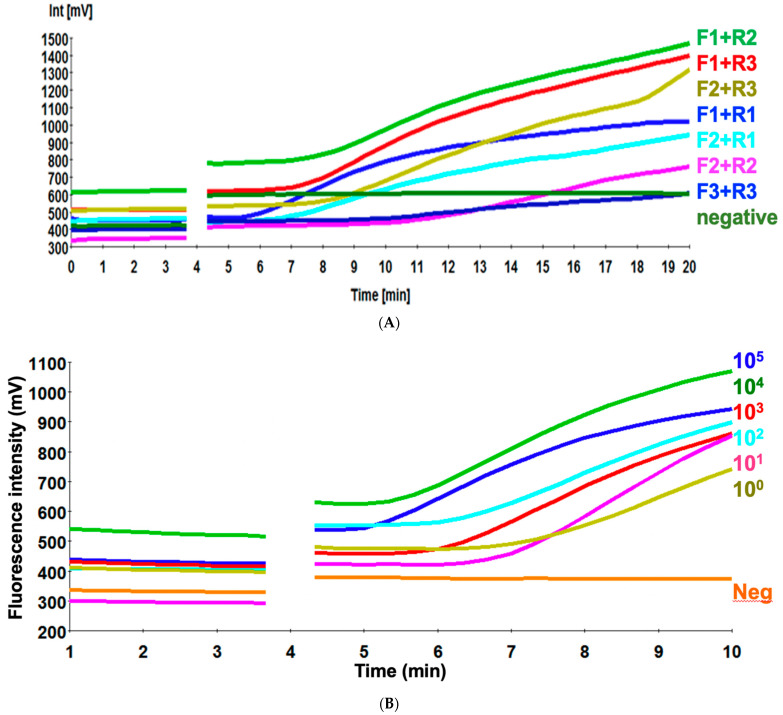
Primer combination testing using 10^5^ EID_50_ per RT-RPA reaction mixture (**A**). Analytical sensitivity of H9 RT-RPA using standard virus of titer 10^5^ EID_50_/mL using a dilution range down to 1 EID_50_/mL (**B**). The combination H9 RPA-F1+ H9 RPA-R1+ H9 exo-probe was detected down to one RNA molecule. The discontinuity at 3 min 50 s due to the necessary mixing step interrupting fluorescence measurement.

**Figure 2 vetsci-08-00134-f002:**
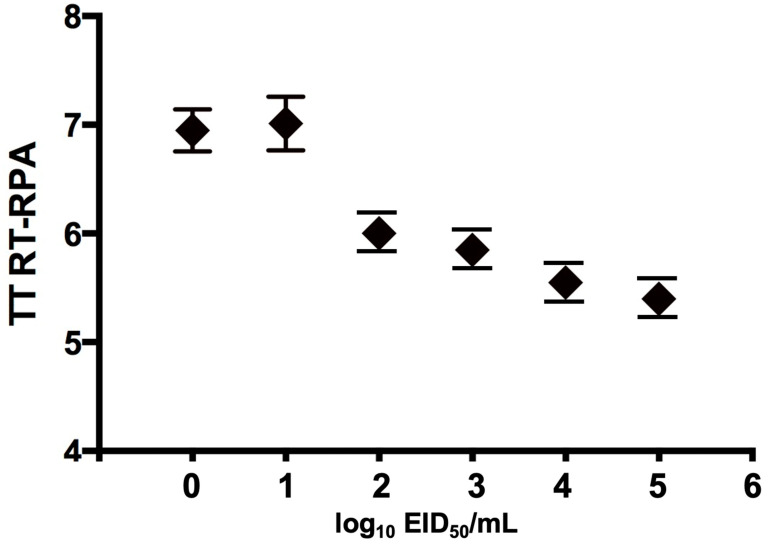
Semi logarithmic regression of the data collected from eight H9 RT-RPA test runs on the viral RNA standard using PRISM Software. H9 RT-RPA results were obtained in 5–8 min. In the H9 RT-RPA assay, 10^0^ EID_50_/mL were detected in 8 out of 8 RT-RPA runs. Rhomboid represents the mean values, and lines are the error bars.

**Figure 3 vetsci-08-00134-f003:**
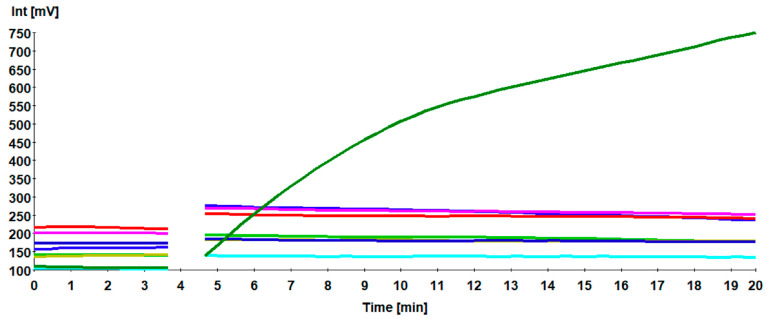
Assay cross reactivity. Only the RNA of aINF H9N2 was amplified (dark green). Nucleic acids of influenza viruses H5N2 (blue), H7N1 (magenta), and ILTV (red), IBV (cyan), NDV (light green), of *Mycoplasma gallisepticum* (khaki) as well as negative control (dark blue) were not detected.

**Figure 4 vetsci-08-00134-f004:**
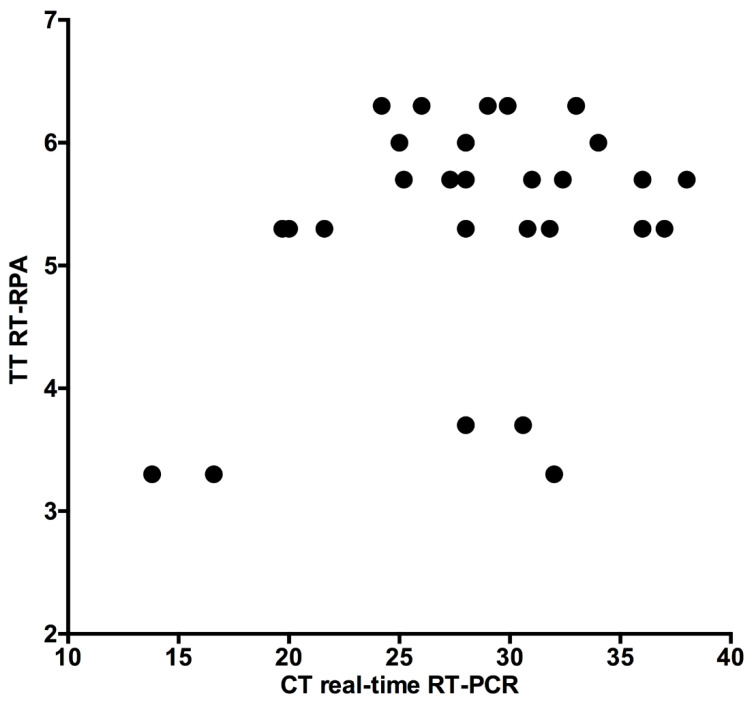
Validation of the RT-RPA assay on the clinical samples compared to real-time RT-PCR. H9 RT-RPA threshold times (TT, Y-axis) and real-time RT-PCR cycle threshold values (CT, X-axis) were compared by PRISM software. R squared (R2) value was 0.12. Independent of the CT values, RT-RPA assay detected all samples within 7 min.

**Table 1 vetsci-08-00134-t001:** Primers and exo-probe used for H9 RT-RPA.

Name	Sequence
H9 RPA-F1	ATGGCTGCAGATAGGGATTCCACTCAAAAGGCAG
H9 RPA-F2	TTCCACTCAAAAGGCAGTTGACAAAATAAC
H9 RPA-F3	TTGGTATGGTTTCCAACATTCAAATGATC
H9 RPA-R1	TAGCAACTCTGCATTATATGCCCATACAT
H9 RPA-R2	TTCACGTTTGCGTCATGCTCATCGAGTGTTTTC
H9 RPA-R3	ATGCTCATCGAGTGTTTTCTGGTTCTCAAG
H9-exo probe	CATGAATTCAGTGAGGTTGAAACTAGAC(BHQ1-dT) (tetrahydrofuran residue) (FAM-dT) ATGATCAATAACA-PH

## Data Availability

All data is included in the manuscript.

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
