# Peer review of "Reverse Transcription Recombinase Polymerase Amplification Assay for Rapid Detection of Avian Influenza Virus H9N2 HA Gene"

_vetsci, 2021, doi:10.3390/vetsci8070134_

Round 1

Reviewer 1 Report

A manuscript (vetsci-1212244) entitled “Reverse Transcription Recombinase Polymerase Amplification Assay for Rapid Detection of Avian Influenza Virus H9N2 HA Gene” by Nahed Yehia et al is well describing the application of reverse transcription recombinase polymerase amplification (RT-RPA) assay for the detection of H9N2 subtype of avian Influenza A virus. As H9N2 avian influenza A virus is endemic in Egypt and has a potential to be a human pathogen, a development of rapid and sensitive virus detection method is very useful to see the spread of infection. Specificity of RT-RPA method was well described by using some other respiratory pathogens such as H5N2 and H7N1 subtypes of avian influenza A virus, NDV, ILT virus, IBV and Mycoplasma gallisepticum. The fact described in this manuscript is worth to be published in the international journal, Veterinary Sciences. However, before accepting this manuscript, several points should be addressed and clarified.

Abstract:

Line 23, A term “H9N2 avian influenza viruses (AIVs)” is not suitable. Influenza A virus, B virus and C virus are abbreviated IAV, IBV and ICV, respectively according to the International Committee on Taxonomy of Viruses (Retrieved 9 March 2021). “H9N2 avian influenza viruses (AIVs)” is correctively shown such as “H9N2 subtype of avian influenza A viruses (aIAVs).

Line 31, An indication of virus quantity, “5 log10 EID50/ml”, is not correct. Mathematically, “x = 10a” is equal to “log x= a”, but “log x” is not equal to “10a”. Thus, the indication should be “105 EID50/ml”. Likewise, lines 121, 162 and 168, and the x-axis footnote of figure 2 should be corrected. The logarithmic indications of x-axis need no correction (Fig. 2).

Line 31, A term “EID50” should be used after the full spelling out. Does this mean 50% embryonating egg infection dose, or embryo infective dose or effective infection dose? Please specify.

Line 41, An explanation about the HA and NA subtypes of influenza A virus are required in “Introduction” part.

Lines 49-50, the words “human-like receptors” and “avian-like receptors” are not sufficient. A supplementary explanation is required such as “human-like receptors (SA, alpha2-3, Gal beta1-4GlcNac)” and “avian-like receptors (SA, alpha2-6 Gal, beta1-4GlcNac)” to specify the target molecules.

Line 54, “H9N2 genotype” is not suitable. H9N2 is one of subtypes of influenza A virus, and not the genotype.

Line 55, Influenza A virus proteins such as HA, NA, M1, M2 and NS1 are used without explanation. Brief explanation about influenza A virus genes or proteins are required.

Materials and Methods:

Lines 79-81; Quantity and quality of H9N2 influenza A viral RNA is one of key issue in this study. However, an explanation in section 2.1. is not enough. Was A/chicken/Egypt/1373Vd/2013 titrated using embryonating hen eggs or embryo, or using something?  Was EID50 value calculated by Reed and Muench method? Was  viral RNA was then extracted from the same sample that was titrated? Please clarify them.

Line 131, “Mycoplasma Gallisepticum” should be “Mycoplasma gallisepticum”.

Lines 139 and 146, “CT” should be defined as cycle threshold at line 139 at first, then use the abbreviated CT at lines 145-146.

Results:

Lines 152-153, Though there are description that Primer pair F1+R1 yielded the best amplification with signals emerging at 5.3 minutes among all the primer combination, but no results was shown anywhere. In addition, primer names are different between the text and Table. In the table all primers are named as H9 RPA-Fx or -Rx, but in the text only Fx or Rx is used. Please unify them throughout the manuscript.

Figure 1, Explain the discontinuity of lines around 4 minutes. An order of EID50 values at the right side of figure does not corelate the fluorescence intensity of each viral dose. What it will be, if Y-axis is plotted by subtraction (DYï¼›Y xmin-Y 1min)?

Lines 176-181, There is no results showing the specificity of H9 RT-RPA assay, but just the description. Add the results obtained through the experiments.

Lines 183-189, No results are shown that thirty clinical samples were detected in about 3-7 min by H9 RT-RPA assay. As a comparative experiment between H9 RT-RPA and real time RT-PCR is important, more detailed descriptions about figure 3 are required. What will be interpreted, if figure 3 is divided by 4 areas (Low TT RT-RPA/Low RT-PCR, Low TT RT-RPA/High RT-PCR, High TT RT-RPA/Low RT-PCR and High TT RT-RPA/High RT-PCR)? In addition, the reason why correlation efficient R2 was low should be discussed much more in Discussion section.

References:

Lines 262, 273, 295, 313, 318, 325 and 330, Page numbers are shown as the shorten form. Please follow the journal guideline.

Line 292, Is the citation of Ref 14 correct? Updated Avian influenza chapter was found in Chapter 3.3.4 OIE p821-841, Terrestrial Manual 2018.

Author Response

Abstract:

Line 23, A term “H9N2 avian influenza viruses (AIVs)” is not suitable. Influenza A virus, B virus and C virus are abbreviated IAV, IBV and ICV, respectively according to the International Committee on Taxonomy of Viruses (Retrieved 9 March 2021). “H9N2 avian influenza viruses (AIVs)” is correctively shown such as “H9N2 subtype of avian influenza A viruses (aIAVs).

Corrected to (IAVs) throughout the manuscript.

Line 31, An indication of virus quantity, “5 log10 EID50/ml”, is not correct. Mathematically, “x = 10a” is equal to “log x= a”, but “log x” is not equal to “10a”. Thus, the indication should be “105 EID50/ml”. Likewise, lines 121, 162 and 168, and the x-axis footnote of figure 2 should be corrected. The logarithmic indications of x-axis need no correction (Fig. 2).

Corrected 

Line 31, A term “EID50” should be used after the full spelling out. Does this mean 50% embryonating egg infection dose, or embryo infective dose or effective infection dose? Please specify.

Full spelling is added. 50 % embryo infective dose(EID50)    

Line 41, An explanation about the HA and NA subtypes of influenza A virus are required in “Introduction” part.  

The required explanation is added to introduction.

Lines 49-50, the words “human-like receptors” and “avian-like receptors” are not sufficient. A supplementary explanation is required such as “human-like receptors (SA, alpha2-3, Gal beta1-4GlcNac)” and “avian-like receptors (SA, alpha2-6 Gal, beta1-4GlcNac)” to specify the target molecules.

The required explanation is clarified.

Line 54, “H9N2 genotype” is not suitable. H9N2 is one of subtypes of influenza A virus, and not the genotype.

Corrected

Line 55, Influenza A virus proteins such as HA, NA, M1, M2 and NS1 are used without explanation. Brief explanation about influenza A virus genes or proteins are required.

Done

Materials and Methods:

Lines 79-81; Quantity and quality of H9N2 influenza A viral RNA is one of key issue in this study. However, an explanation in section 2.1. is not enough. Was A/chicken/Egypt/1373Vd/2013 titrated using embryonating hen eggs or embryo, or using something?  Was EID50 value calculated by Reed and Muench method? Was viral RNA was then extracted from the same sample that was titrated? Please clarify them.

Done

Line 131, “Mycoplasma Gallisepticum” should be “Mycoplasma gallisepticum”.

Corrected

Lines 139 and 146, “CT” should be defined as cycle threshold at line 139 at first, then use the abbreviated CT at lines 145-146.

CT is defined

Results:

Lines 152-153, Though there are description that Primer pair F1+R1 yielded the best amplification with signals emerging at 5.3 minutes among all the primer combination, but no results was shown anywhere. In addition, primer names are different between the text and Table. In the table all primers are named as H9 RPA-Fx or -Rx, but in the text only Fx or Rx is used. Please unify them throughout the manuscript.

Various Primer combination results were displayed in figure 1A. Primer names are unified throughout the text to be same as the table 1.

Figure 1, Explain the discontinuity of lines around 4 minutes. An order of EID50 values at the right side of figure does not corelate the fluorescence intensity of each viral dose. What it will be, if Y-axis is plotted by subtraction (DYï¼›xmin-Y 1min)?

The discontinuity at 4 min due to the mixing of the samples during the test according to method. The Figure was updated accordingly.

Lines 176-181, There is no results showing the specificity of H9 RT-RPA assay, but just the description. Add the results obtained through the experiments.

Figure 3 was added to cover this point.

Lines 183-189, No results are shown that thirty clinical samples were detected in about 3-7 min by H9 RT-RPA assay. As a comparative experiment between H9 RT-RPA and real time RT-PCR is important, more detailed descriptions about figure 3 are required. What will be interpreted, if figure 3 is divided by 4 areas (Low TT RT-RPA/Low RT-PCR, Low TT RT-RPA/High RT-PCR, High TT RT-RPA/Low RT-PCR and High TT RT-RPA/High RT-PCR)? In addition, the reason why correlation efficient R2 was low should be discussed much more in Discussion section.

Many thanks for the suggestion, figure 3 and the discussion were updated accordingly. Since the RPA detected all PCR positive samples, we did not include a classification of the CT values, but it clears from the graph that samples with various CT were included.

References:

Lines 262, 273, 295, 313, 318, 325 and 330, Page numbers are shown as the shorten form. Please follow the journal guideline.

Corrected following the journal guideline and highlighted.

Line 292, Is the citation of Ref 14 correct? Updated Avian influenza chapter was found in Chapter 3.3.4 OIE p821-841, Terrestrial Manual 2018.

updated

Reviewer 2 Report

General comments

1. The manuscript describe a novel assay for detecting H9 virus circulating in Egypt and may not be valid against virus circulating elsewhere. 

2. Diagnostic time reduced by this assay only impact one of many steps from sample collection, transportation, RNA extraction. 

Introduction

3. Please add details on H9 subclades detected elsewhere compared to those in Egypt 

4. Please add review of other rapid molecular diagnostic for comparison i.e. iiPCR (insulated isothermal PCR)

Materials and methods

5. Line 116 - please specify the level of threshold used

5. Line 119 - the heading should be "Analytical sensitivity" (see https://www.ncbi.nlm.nih.gov/pmc/articles/PMC2901657/pdf/0074-09.pdf) 

6. Line 121 - Please clarify if all eight replicates were run on the same day

7. Line 125 - Please add description (variables included in the model) and provide reference for the semi-log regression

8. Line 127 - the heading should be "Analytical specificity" 

9. Line 135 - the heading should be "Accuracy"

10. The number of sample for accuracy determination is low (30 positive, 12 negative compared to CLSI recommended 50 each - see reference above).

11.  Linear regression imply cause-effect model. Pearson correlation should be used to determine the level of correlation between two normally distributed variables. Otherwise use Spearman rank correlation. Please also add description (variables in the model) and reference for statistical method used

12. Figure 1 - why are the lines disconnected at 4 minutes? Please explain or make them continue. Please also add threshold line (horizontal).

13. Figure 2 - Please add label what are the markers and line indicate (estimated TT from regression model?)

Discussion

14. Line 218 - If probe does not count, oligonucleotides can be replaced with "primers" 

15. Line 223 - Replace "coincidence rate" with "agreement"

16. Line 240 - replace "low" with "law"

Author Response

General comments

Introduction

3. Please add details on H9 subclades detected elsewhere compared to those in Egypt 

Sentences to cover this point was added in the introduction.

4. Please add review of other rapid molecular diagnostic for comparison i.e. iiPCR (insulated isothermal PCR).

Other rapid diagnostic is already compared and discussed in discussion such as the loop-mediated isothermal amplification (RT-LAMP) assay and RT-RPA-FLD assay. We added review on iiPCR too. RT-PCR still the gold standard diagnostic molecular method so, all developed rapid methods should be compared to RT-PCR.So, we compared it to RT-PCR in introduction while in discussion we discussed how our developed protocol is faster than other rapid diagnostic methods and in the same time sensitive and specific as RT-PCR.

Materials and methods

5. Line 116 - please specify the level of threshold used

Statistical analysis method was added in the Material and methods.

 5. Line 119 - the heading should be "Analytical sensitivity" (see https://www.ncbi.nlm.nih.gov/pmc/articles/PMC2901657/pdf/0074-09.pdf) 

Corrected

6. Line 121 - Please clarify if all eight replicates were run on the same day

NO. They were done in different days.

7. Line 125 - Please add description (variables included in the model) and provide reference for the semi-log regression

Statistical analysis method was added in the Material and methods.

8. Line 127 - the heading should be "Analytical specificity" 

Corrected

9. Line 135 - the heading should be "Accuracy"

Corrected

10. The number of sample for accuracy determination is low (30 positive, 12 negative compared to CLSI recommended 50 each - see reference above).

Many thanks for the suggestion. This was the number of samples that was provided by the reference lab. Off course we can collect more sample, but a special approval was necessary.

11.  Linear regression imply cause-effect model. Pearson correlation should be used to determine the level of correlation between two normally distributed variables. Otherwise use Spearman rank correlation. Please also add description (variables in the model) and reference for statistical method used

A section of the statistical methods used in the study is included in the Material and methods.

12. Figure 1 - why are the lines disconnected at 4 minutes? Please explain or make them continue. Please also add threshold line (horizontal).

Due to the mixing occurred during experiment as standard method of RPA kit.

13. Figure 2 - Please add label what are the markers and line indicate (estimated TT from regression model?)

Figure caption was updated according to the reviewer suggestion. 

Discussion

14. Line 218 - If probe does not count, oligonucleotides can be replaced with "primers" 

Many thanks for the suggestion. Theword “primers” was added instead of “oligonucleotides”.

15. Line 223 - Replace "coincidence rate" with "agreement"

corrected

16. Line 240 - replace "low" with "law"

corrected

Reviewer 3 Report

The authors in the present study developed a Reverse Transcription Recombinase Polymerase Amplification Assay for Rapid Detection of Avian Influenza Virus H9N2 HA Gene. They found that the method is sensitive, accurate, rapid, specific, and could be applied easily at field.

The manuscript to me is, in general, clearly written. The science and technical execution of the study is of good quality. The study is solid and the data, in general, support the conclusions. The theory, logic, and experimental design are easy to follow and in general, make sense. 

However, some clarification is necessary.

Comments

How about the difference with other studies like: https://pubmed.ncbi.nlm.nih.gov/30403438/

How about the cost compared with PCR?

I don't agree with the statement that it could be applicable at field conditions because preparation for these experiments must be in the lab.

Line 51+ 52 and 57 are contradicting.

PCR preparations are missing.

Adjust Figure 1.

Author Response

How about the difference with other studies like: https://pubmed.ncbi.nlm.nih.gov/30403438/

We already cited this article (reference #21) and compared our probe-based RPA assay to RT-RPA-FLD assaythat developed in this cited article. Our probe-based RPA assay is faster (7 min) than RT-RPA-FLD assay because it can generate real-time amplification result in less than 20 min without an additional process after amplification. In addition, the clinical performance of H9 RT-RPA is slightly better than H9 RT-RPA-FLD.

How about the cost compared with PCR?

It is the same cost. We have indicated this in one of our previous publications.

I don't agree with the statement that it could be applicable at field conditions because preparation for these experiments must be in the lab.

Many thanks for the comment, but we have applied this technology in field settings, e.g. Ebola in Guinea and for FMDV in Egypt

Line 51+ 52 and 57 are contradicting.

In line 51 and 52, we stated that H9N2 has emerged and become endemic in Egypt since 2010 and causing serious economic impact when combined with secondary infections. In line 57, we discussed the problem of LPAIVs as H9N2 is low pathogenic (LPAI) but asymptomatically infected poultry shed and spread the virus that possibly generating new sub- and genotypes of AI, thus potentially lead to new influenza pandemics. So, however it is low pathogenic but still serious to poultry industry and public health.

PCR preparations are missing.

 It already mentioned in detail in materials and methods under subtitle “2.5. Optimization of H9 RT-RPA conditions”

Adjust Figure 1.

Corrected

Round 2

Reviewer 1 Report

Revised manuscript (vetsci-1212244) entitled “Reverse Transcription Recombinase Polymerase Amplification Assay for Rapid Detection of Avian Influenza Virus H9N2 HA Gene” by Nahed Yehia et al is almost properly revised according to the reviewer’s suggestions and comments. Details of this manuscript are shown clearer by this revision and these efforts made the key issue of this manuscript more understandable. The manuscript has reached the level that needs to clear the major concern. However, this reviewer thinks this manuscript still has minor points that should be clarified before accepting.

Introduction:

Lines 41-43, An explanation about the HA and NA subtypes of influenza A virus was properly added as suggested. However, brief explanation about influenza A virus genes or proteins are not shown well, but simply showing M and NS full-spelling. An explanation like “The Influenza A virus (IAV)  has eight RNA segments as a genome. Each one is called RNA polymerase basic subunit (PB)2, PB1, RNA polymerase acidic subunit (PA), hemagglutinin (HA), nucleoprotein (NP), neuraminidase (NA) and matrix (M) and non-structural (NS) segment.” is required before tolking M and NS genes.

  • Line 85-86, An explanation about the HA and NA subtypes of influenza A virus are required in “Introduction” part.  

   Corrected  

  • Line 55, Influenza A virus proteins such as HA, NA, M1, M2 and NS1 are used without explanation. Brief explanation about influenza A virus genes or proteins are required.

   Done

Lines 251-252, Related to above comments, the author modified sentences, but an sentence matrix protein-1 (M1), -2 (M2) and non-structural (NS) protein sequences” should be matrix protein-1 (M1), -2 (M2) and NS protein gene sequences”.

Materials and Methods:

2.2. Generation of RNA standard

Lines 529-533, As sentence was revised by the author, but this sentence is still not clear. I suppose that “The standard RNA was extracted from Egyptian H9N2 reference strain [A/chicken/Egypt/1373Vd/2013(H9N2), GenBank accession number KJ781216] that was titrated using specific pathogen free (SPF) embryonated chicken eggs (ECE). Virus titer shown as EID50 (50% embryo infective dose)/ml was calculated using the Reed and Muench method as previously described [25]”. Please clarify the methods.

  • Lines 79-81; Quantity and quality of H9N2 influenza A viral RNA is one of key issue in this study. However, an explanation in section 2.1. is not enough. Was A/chicken/Egypt/1373Vd/2013 titrated using embryonating hen eggs or embryo, or using something?  Was EID50value calculated by Reed and Muench method? Was viral RNA was then extracted from the same sample that was titrated? Please clarify them. 

   Done

2.3. Viral RNA extraction

Lines 537-539; Meaning of newly inserted sentence, ”Samples were either allantoic fluid or phosphate-buffered saline (PBS) suspensions containing tracheal swabs of a total volume of 200 µl.”, is unclear. Does it mean that “Two hundred µl of allantoic fluid or tracheal swabs suspended with phosphate-buffered saline (PBS) were used as the sample to be measured.”? Please clarify the methods.

2.7. H9 RT-RPA Analytical specificity

Lines 602-603; Add the “virus” between “infectious laryngotracheitis” and “(ILTV)”, and also between “infectious bronchitis” and “(IBV)”.

Results:

New Figure 3; Several color lines are shown in figure 3, but no explanation except green line. Even if they were not amplified, careful handling of results is required.

  • Lines 176-181, There is no results showing the specificity of H9 RT-RPA assay, but just the description. Add the results obtained through the experiments.

   Figure 3 was added to cover this point.

Author Response

Many thanks for your comments. We have updated our manuscript accordingly. 

Reviewer 2 Report

The manuscript is significantly improved, but require editing before being accepted.

Author Response

Thanks for taking the time to review our study.